# Recent Advances in Resonator Networks for Neuro-Symbolic Computing

**Alpha Renner**                                                                A.RENNER@FZ-JUELICH.DE
*Forschungszentrum Jülich, Germany*

**Christopher Kymn**                                                            CJKYMN@BERKELEY.EDU
*UC Berkeley, USA*

**E. Paxon Frady**                                                             E.PAXON.FRADY@INTEL.COM
*Intel, USA*

**Friedrich T. Sommer**                                                        FSOMMER@BERKELEY.EDU
*UC Berkeley, USA*

**Editors:** Leilani H. Gilpin, Eleonora Giunchiglia, Pascal Hitzler, and Emile van Krieken

## Abstract

As an alternative to hybrid neuro-symbolic approaches for combining the strengths of Neural Networks and symbolic AI, Vector Symbolic Architectures (VSAs) provide a seamless framework for robust parallel computing with transparency. A product-like dyadic vector operation in VSAs enables variable binding and the encoding of data structures by decomposable distributed representations. One critical, long-standing problem in VSA-style neuro-symbolic computing was how to decompose a representation of bound variables. Given the encoding schemes of the variables, pattern matching and unbinding involves vector factorization, that is, searching large combinatorial spaces of vector products. By interleaving binding operations with attractor network dynamics, Resonator Networks can efficiently solve this problem.

Recent advances demonstrate the versatility and potential of Resonator Networks. Here, we present four key developments published in the past year: a hierarchical resonator network handling non-commutative transformations in composed visual scenes, a neuromorphic hardware implementation, applications to visual odometry, and insights into cognitive map formation in hippocampal-entorhinal circuits. These studies offer theoretical insights and practical applications for neuro-symbolic parallel computing. In the future, incorporating learning mechanisms for the underlying generative models in Resonator Networks offers a promising path toward compositional reasoning in neuro-symbolic AI systems.

Vector Symbolic Architectures (VSAs) enable neuro-symbolic computing through high-dimensional vector operations: superposition, binding, and permutation (Plate, 1995; Kanerva, 1996; Gayler and Wales, 1998; Frady et al., 2020b). The multiplicative binding operation creates composite representations, but recovering individual factors from bound vectors poses a combinatorial factorization challenge (Frady et al., 2020a; Kent et al., 2020). Resonator Networks address this through iterative dynamics interleaving binding with pattern completion (Frady et al., 2020a; Kent et al., 2020), leveraging superposition to search multiple potential factorizations simultaneously. Recent work has significantly extended the versatility and applicability of resonator networks, with several key developments:

### Visual Scene Understanding

Hierarchical Resonator Networks (HRNs) (Renner et al., 2024b) now handle non-commutative geometric transformations in visual scenes encoded by fractional power encoding (Plate, 1995; Frady et al., 2018; Komer et al., 2019; Frady et al., 2021). This is achieved through partitioned architectures simultaneously operating in multiple reference frames (Renner et al., 2024b), outperforming deep learning with limited data on artificial 2d scenes. Kymn et al. (2024c) show that augmenting resonator networks with convolutional sparse coding improves pattern separation, accuracy, and convergence time. Hersche et al. (2025); Karunaratne et al. (2024) further address the resonator network's operational capacity by exploring the benefits of noise and sparsity.

### Visual Odometry Application

Real-world applications in motion estimation (visual odometry) using event-based cameras showcases the resonator networks practical utility (Renner et al., 2024a). The system represents images as sums of products of VSA vectors and uses hierarchical resonator networks to estimate camera motion. Performance evaluation shows competitive performance requiring fewer computational resources than large neural networks.

### Neuromorphic Implementation

Addressing computational efficiency on classical hardware, neuromorphic hardware implementations of the resonator achieve in-memory factorization of holographic representations (Renner et al., 2024b; Langenegger et al., 2023; Wan et al., 2024). Renner et al. (2024b) uses complex-valued Fourier Holographic Reduced Representations (Plate, 1995), efficiently encoding complex phase as spike timing. They can also be implemented by continuous-time oscillator circuits (Kymn et al., 2025), which improve representational capacity.

### Understanding cognitive maps

Resonator networks can also provide principled models of cognitive computations in the brain. For example, Kymn et al. (2024a) models attractor dynamics in the hippocampus and entorhinal cortex in part based on resonator networks. Such a system yields insights into the high-capacity and likely compositional nature of grid-cell codes, and improvements for positional encodings in VSA and resonator networks (Kymn et al., 2024b). Furthermore, building on previous work that links VSAs to grid-cells (Frady et al., 2018; Komer et al., 2019), a grid-cell-inspired structured vector algebra (GC-VSA) (Krausse et al., 2025) applies resonator networks to read out maps and episodic memory representations formed by binding together timing, location and semantic information.

## Discussion and Future Directions

These advances position resonator networks as a bridge between symbolic and neural computation, addressing fundamental limitations of neural networks in compositional generalization, interpretability and efficiency. Interestingly, similar principles may be employed

in biological systems (Kymn et al., 2024a; Krausse et al., 2025), where inference requires recurrent dynamics to uncover the factors of variation.

Future directions include i) scaling the resonator network to parse more complicated structures, such as in the visual disentanglement literature, ii) incorporating techniques from energy-based models to learn more powerful data-driven generative models, and iii) improving performance on combinatorial optimization problems such as subset sum (Kleyko et al., 2022; Kymn et al., 2024b, 2025).

The convergence of theoretical advances (Frady et al., 2020a; Kent et al., 2020; Kleyko et al., 2023; Renner et al., 2024b; Kymn et al., 2024b; Karunaratne et al., 2024; Hersche et al., 2025), practical applications (Renner et al., 2024b,a; Kymn et al., 2024c), and biological insights (Kymn et al., 2024a,b; Krausse et al., 2025) establishes resonator networks as a promising candidate for next-generation interpretable neuro-symbolic AI systems capable of compositional reasoning.

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
