# OpenReview forum: "Recent Advances in Resonator Networks for Neurosymbolic Computing"
_nesyconf.org/NeSy/2025/Conference_Phase_2 — NeSy 2025 - Phase 2 Poster_

### Official Review · Reviewer_csf4 · 2025-07-02
**Resonator Networs for Vector Symbolic Architectures and Applications**

**Rating:** 7
**Confidence:** 3

**Review:**

This article addresses a key challenge in Vector Symbolic Architectures (VSAs): how to unbind individual components from bound high-dimensional vectors, a task that normally requires a difficult combinatorial search.
Resonator Networks solve this by combining vector binding with recurrent attractor dynamics, allowing the system to iteratively converge on the correct factors through distributed pattern matching.
Building on this, the paper highlights four recent advances:
- A hierarchical resonator network that interprets visual scenes with complex object transformations.
- A visual odometry system using event-based cameras that tracks motion efficiently with fewer resources.
- Hardware implementations that perform vector operations in-memory using neuromorphic circuits.
- Cognitive models that simulate how the brain builds spatial maps and episodic memories.
The collection of work show how resonator networks are evolving into practical tools for neuro-symbolic reasoning, with future directions focused on learning more complex structures and solving harder combinatorial tasks.

Minor comments:
- The reader would benefit from a comparison between resonator-based VSA to other neurosymbolic approaches like LTNs or DeepProbLog.

**Anonymity:**

Remain anonymous

---

### Official Review · Reviewer_cR7u · 2025-07-03
**Unclear connection between hypotheses and theses - "Recent Advances in Resonator Networks for Neurosymbolic Computing"**

**Rating:** 3
**Confidence:** 4

**Review:**

In this paper, the authors briefly mention that Resonator Networks have been successfully applied to tasks such as motion estimation (visual odometry) and modeling of brain functions. In the discussion section, they highlight scalability and integration with GenAI frameworks as important future directions, and suggest that Resonator Networks are promising candidates for compositional reasoning.

However, I find that the extended abstract lacks a coherent narrative. It remains unclear what underlying properties of Resonator Networks make them suitable for such a diverse set of tasks. Furthermore, the claim that these networks are well-suited for complex reasoning is not sufficiently substantiated within the current text. A more detailed explanation or empirical support would strengthen the argument and help the reader understand the broader potential of this approach.

**Anonymity:**

Remain anonymous

---

### Official Review · Reviewer_u7h9 · 2025-07-10
**Survey on recent work in resonator network.**

**Rating:** 7
**Confidence:** 3

**Review:**

Recent Advances in Resonator Networks for Neuro-Symbolic Computing provides a synopsis of four key developments in resonator networks over the past year: hierarchical resonator networks for non-commutative scene understanding, neuromorphic hardware implementations, visual odometry applications, and insights into cognitive map formation.

The paper presents a concise synthesis of four major strands of recent work on resonator networks. It has a clear structure, with well-defined sections on theory, hardware, applications, and neuroscience, defining a strong linkage between neuro-symbolic computing and biological inspiration.

The paper assumes the reader's familiarity with VSAs or resonator networks. Key algorithms are described at a high level without detailed pseudocode or equations.

**Anonymity:**

Remain anonymous